# The Effect of Exercise Training and Royal Jelly on Hippocampal Cannabinoid-1-Receptors and Pain Threshold in Experimental Autoimmune Encephalomyelitis in Rats as Animal Model of Multiple Sclerosis

**DOI:** 10.3390/nu14194119

**Published:** 2022-10-03

**Authors:** Maryam Kheirdeh, Maryam Koushkie Jahromi, Annette Beatrix Brühl, Serge Brand

**Affiliations:** 1Department of Sport Sciences, School of Education and Psychology, Shiraz University, Shiraz 7194684334, Iran; 2Center for Affective, Stress- and Sleep Disorders (ZASS), Psychiatric Clinics (UPK), University of Basel, 4002 Basel, Switzerland; 3Sleep Disorders Research Center, Kermanshah University of Medical Sciences, Kermanshah 6714869914, Iran; 4Substance Abuse Prevention Research Center, Kermanshah University of Medical Sciences, Kermanshah 6714869914, Iran; 5Department of Sport, Exercise, and Health, Division of Sport Science and Psychosocial Health, University of Basel, 4052 Basel, Switzerland; 6Department of Psychiatry, School of Medicine, Tehran University of Medical Sciences, Tehran 1417613151, Iran

**Keywords:** endurance training, royal jelly, pain, cannabinoid-1-receptors (CB1R), experimental autoimmune encephalomyelitis

## Abstract

Cannabinoid-1-receptors (CB1R) are therapeutic targets for both the treatment of autoimmune diseases, such as multiple sclerosis (MS), and some related symptoms such as pain. The aim of this study was to evaluate the effect of aerobic training and two dosages of royal jelly (RJ) on hippocampal CB1R and pain threshold (PT) in an experimental autoimmune encephalomyelitis (EAE) model. To this end, 56 female Sprague-Dawley rats with EAE were randomly assigned to one of the following eight conditions: (1) EAE; (2) sham; (3) 50 mg/kg RJ (RJ50); (4) 100 mg/kg RJ (RJ100); (5) exercise training (ET); (6) ET + RJ50; (7) ET + RJ100; and (8) not EAE or healthy control (HC). Endurance training was performed for five weeks, four sessions per week at a speed of 11–15 m/min for 30 min, and RJ was injected peritoneally at doses of 50 and 100 mg/kg/day). One-way analysis of variance and Tukey’s post hoc tests were performed to identify group-related differences in pain threshold (PT) and CB1R gene expression. Endurance training had no significant effect on PT and hippocampal CB1R in rats with EAE. CB1R gene expression levels in the RJ100 group were higher than in the EAE group. Further, PT levels in the ETRJ50 and ETRJ100 groups were higher than in the EAE group. The combination of ET and RJ50 had a higher impact on PT and CB1R, when compared to the ET and RJ50 alone. Next, there was a dose-response between RJ-induced CB1R gene expression and RJ dosages: higher dosages of RJ increased the CB1R gene expression. The overall results suggest that the combination of ET and increasing RJ dosages improved pain threshold probably related to CB1R in an EAE model, while this was not observed for ET or RJ alone.

## 1. Introduction

Multiple sclerosis (MS) is one of the most common autoimmune and neurodegenerative diseases [1,2]. Prevalence and incidence rates vary between two per 100,000 in Japan to greater than 100 per 100,000 in Northern Europe and North America [3]. 

MS is considered an inflammatory and oxidative disease characterized by myelin destruction, sensory and motor dysfunction, and cognitive impairment, and chronic pain [4]. As such, MS imposes a heavy burden in terms of suffering and loss of function.

At the initial stage of MS, it appears that myelin degradation, immune system disorders, neuronal apoptosis, and oxidative stress unfavorably impact on brain tissue volume, decrease cerebral blood flow, and develop inflammatory responses [4]. Importantly, such pathways lead to the destruction of sensors in general and to the destruction of pain sensors specifically [4]. In addition, inflammatory disorders such as MS destroy parts of the central nucleus of the amygdala (CeA), which plays a major role in the neuronal elaboration of the pain process [5,6]. More specifically, inflammatory disorders such as MS destroy the nucleus of the amygdala (CeA), and negatively impact on the circulating levels of opioids, and cannabinoids such as cannabinoids type 1 (CB1) and its receptor (CB1R). Relatedly, inflammatory disorders such as MS negatively impact on the neurotransmitters such as morphine reward and morphine antinociception, which are highly involved in the neuronal elaboration of pain [5,6].

In treating symptoms of MS and its progress, medication-based [1,2,7,8] and non-medication-based [9] treatments show encouraging results. For non-medication-based interventions, in the present study, we focused on interventions of regular physical activity and exercising [10,11,12], and royal jelly as an important anti-inflammatory factor [13,14].

As regards regular physical activity and exercising, there is extant research that regular physical activity and exercising favorably impacts on a broad range of MS-related symptoms, such as paresthesia as a proxy of pain elaboration [15,16,17]. Exercise seems to have beneficial effects on neurodegenerative diseases and is useful for improving neurotransmitters [18], oxidative factors, as well as cognitive function [19]. In addition, it appears that regular physical activity increases CB receptors and improves their function by reducing oxidative stress and inflammation, and increasing neurotrophins [20]. However, to our knowledge, the neuronal mechanisms of pain regulation in MS have not been investigated so far. Given this, the aim of the present study was to investigate the effect of regular exercising on possible neuronal pathways for pain elaboration as CB receptors in an animal model of MS. 

Experimental autoimmune encephalomyelitis (EAE) is the most common type of animal model inducing MS (MS), and increase of inflammatory factors causes demyelination or MS [21]. EAE’s validity as a good model of MS has been shown by various studies [22]. Endurance training has been shown to reduce inflammatory factors, reduce apoptosis, and increase brain-derived neurotrophic factor in an EAE model [23]. According to a review study, exercise enhances CB activity by improving neurotransmitters and neuronal plasticity pathways, and improves cognitive function, memory, as well as learning in degenerative disorders [24]. 

Besides exercise, royal jelly (RJ), due to its antioxidant anti-inflammatory and anti-apoptotic effects, is used in the treatment of many diseases, including neurodegenerative diseases [25]. Studies show that RJ is associated with modulating neurotransmitters involved in depression and anxiety and improving chronic pain neuroplasticity [18,25]. In this regard, eight weeks of RJ consumption improved pain threshold in Alzheimer’s rats [26]. Despite reports of the beneficial effects of RJ on improving nervous system function either in separate or in combination with exercise, the cannabinoid receptor and its dependent analgesic effects regarding these two interventions are still not well understood. However, to our knowledge, the neuronal mechanisms of pain regulation in MS have not been investigated so far. Therefore, the aim of the present study was to evaluate the effect of aerobic exercise training (ET) with royal jelly consumption in two doses on CB1R in hippocampal tissue and pain threshold (PT) in an EAE model. 

## 2. Methods

### 2.1. Animals and Implementation of Research Design

In this experimental study, 56 female Sprague-Dawley rats with an approximate age of 8–10 weeks and a weight of 200–220 gram were purchased and transferred to the animal sports physiology laboratory. The rats were maintained in the laboratory for one week to adapt to environmental conditions. During the animal research period, the animals were kept in transparent polycarbonate cages in the standard conditions of light (12–12 h of light−darkness cycle), temperature of 22–24 °C, and humidity of 55%. They had free access to water and food during the whole period, and grated sterile wood was used to absorb their urine. In addition, staff tried to keep the laboratory environment free of any extra noise or stress. During the experiment, only the executor attended the animal house at specific times and performed animal interventions. All study procedures were performed according to the Helsinki agreement considering ethical principles of working with animals. The proposal and study procedures were also approved under the ethical principles of working with animals of Shiraz University and supervision of the department of sport sciences. This study was registered and approved by the code of 566250 at 19 June 2021.

### 2.2. Induction of EAE Disease

In order to induce EAE, 20 guinea pigs were prepared from the Pasteur Institute of Iran; then, the spinal cord tissue of guinea pigs (as an antigen for adjuvant effect on the nervous system) was extracted after anesthesia using ketamine and xylazine and immediately placed in a nitrogen tank. The next day, the spinal cord was incubated into a nitrogen-filled mortar. To obtain a homogeneous solution of spinal cord tissue, an equal volume of spinal cord was mixed with normal saline and placed in a shaker at 5 °C until completely homogenization.

This solution was then combined in a 1:1 ratio with complete Freund’s adjuvant (CFA) to form an emulsion. In order to prepare an injectable suspension, two glasses of syringes were used, being connected by a steel interface. One of the syringes contained the homogenized guinea pig’s brain and spinal cord and the other syringe contained the same volume of complete Freund’s adjuvant (CFA); the solution was mixed in equal proportions and its color was made uniform and whitened using a shaker. After rats’ complete anesthesia with ketamine and xylazine, 400 μL of the antigen and adjuvant mixture was injected subcutaneously in the back and 100 μL into the cushion area of each animal with a 25G needle. 

During the daily assessment of the disease process and the condition of animals, in the case of appearance of movement disorders and psychological disorders, rats were noted as EAE-affected.

It should be noted that the disease scale was set as follows:

Zero: no disease, 1: tail movement disorder, 2: tail paralysis, 3: gait disorder, 4: one-leg paralysis, 5: two-leg paralysis, 6: hands and legs paralysis, and 7: death [27,28]. In addition, given to the research requirement for animals’ minimal daily activities, rats in 2–5 scales were included and rats on scales 1, 6, and 7 were typically excluded from the study. Histopathological analyses of hippocampus that proved EAE effects are presented in Figure 1. 

After proving of EAE according to the scales, 49 rats with EAE were divided into seven intervention-matched groups according to EAE scales (EAE score in all related groups: 3.5 ± 1.5) including: (1) EAE, (2) sham, (3) 50 mg/kg RJ (RJ50), (4) 100 mg/kg RJ (RJ100), (5) ET, (6) ET + RJ50, and (7) ET + RJ100. Seven healthy rats were placed in the healthy control (HC) group to evaluate the effects of EAE induction on the research variables. Rats in the sham group received normal saline daily for 5 weeks, the RJ groups received daily doses of royal jelly (dissolved in normal saline) daily for 5 weeks [29], and rats in the ET groups performed programmed aerobic (endurance) training [30,31]. 

### 2.3. Training Protocol

Endurance training began approximately 10 days after the induction of the EAE model. First, before the main training protocol and after induction of EAE, rats were familiarized with the treadmill for one week, with every session lasting 5 to 25 min at a speed of 6 m/s and a slope of 11 degrees [30,31]. Next, once rats were familiarized with the treadmill in the main endurance training protocol, they ran on the treadmill at a speed of 11 m/min for 25–35 min, five sessions per week for 5 weeks.

The duration of the training was 25 min in the first week, but due to the progressive motor disorder in rats with EAE, 2 min per week was added to the duration, so that the training time reached 35 min in the fifth week. One of the reasons for choosing this training protocol was the neuroprotective effects of this type of training in rats with cognitive impairments in Parkinson’s and EAE rats [30,31]. 

### 2.4. Royal Jelly Supplementation

Royal jelly was used with doses of 100 and 50 mg/kg during five weeks. RJ was provided from Marvdasht Agricultural Jihad Center and dissolved in normal saline daily as required; then, it was injected peritoneally into rats [29]. 

### 2.5. Pain Threshold (PT) Evaluation

Pain threshold (PT) was measured using a hot plate analgesia device 24 h after the last session of training protocol. This is a device for measuring acute pain caused by heat. This device has a flat and heating plate made of steel which is controlled by electric current and a precise thermostat. To evaluate PT, the rats were placed on a screen with normal temperature for 3 min one day before the main test to be familiarized with the environment. Then, during the next day, the rats were placed on a hot plate with a temperature of 52.8 °C. During this time, the interval time between the rats being placed on the screen and the first reaction of animals to the temperature, such as raising the leg, jumping, and vibration of foot while lifting, was recorded by a chronometer. This time was considered as the pain threshold. Please note that the maximum test time for each animal was considered 60 s in order to prevent tissue damage [26].

### 2.6. Measurement of CB1R

Forty-eight hours after the last training session, rats were anesthetized using a combination of ketamine (50 mg/kg) and xylazine (15 mg/kg). After ensuring complete anesthesia, the upper part of the skull was removed using a cutter and after observing the brain tissue, it was carefully extracted by specialists. Then, to prevent tissue destruction in the environment, the hippocampus was carefully separated by cutting the lateral lobes and brain tissue. Next, to prevent rapid freezing and freezing water molecules damaging the hippocampus, the tissue was first kept in −20 °C refrigerator for three hours and was then transferred to a −70 °C freezer, and then in the next step, it was tested to measure CB1R gene expression levels using real-time PCR.

In order to measure the CB1R gene expression levels, 50 mg of tissue from the hippocampus was taken to extract RNA. The RNA extraction was performed according to the protocol of the manufacturer (Qiagen, Hilden, Germany); also, to ensure the quality of RNA, agarose gel electrophoresis (Sigma, Aldrich, Saint Louis, MO, USA) was used with light absorption property at 260 nm. Then, RNA with suitable concentration was used for cDNA synthesis based on the protocol of the manufacturer of cDNA fermentase synthesis kit (K1621).

To perform the reverse transcription reaction, the cDNA was mixed with the designed primers (Table 1) by obtaining information from the PUBMED site; also, to determine the efficiency and specificity of the primers, the software available on the NCBI site was used. In addition, the TBP internal control gene was used to measure CB1R gene expression levels. After the quantitative real-time polymerase chain reaction (QRT-PCR) was completed and the samples reached the cycle threshold, the formula 2^−ΔΔCT^ was used to quantify the ratio of the given CB1R gene to the reference gene.

### 2.7. Data Analysis Procedure

The Shapiro−Wilk test was used to evaluate the normality of the findings. One-way analysis of variance (ANOVA) was used to examine the between-group differences, and Tukey’s post hoc test in Graphpad Prism 8.3.6 software was used to determine the place of differences between groups (*p ≤* 0.05). 

## 3. Results

The weights of rats in eight groups are presented in Figure 2. The weights of animals in all groups except RJ100 increased post- compared to pretreatment (*p* ≤ 0.05). Posttreatment, weights of animals in RJ50, RJ100, ET, and ET + RJ100 groups were lower than those of the HC group (*p* ≤ 0.05), while there was no significant difference between EAE and sham groups and the HC group (*p* > 0.05). 

The results of one-way analysis of variance showed a significant difference in CB1R (*p* = 0.001) and PT (*p* = 0.001) levels in the research groups.

The results of Tukey’s post hoc test showed that PT levels in the EAE-induced groups (EAE, Sh, RJ50, RJ100, ET, ET + RJ50, ET + RJ100) were significantly lower than those in the HC group (*p* = 0.001). PT was not significantly different between the sham and EAE groups (*p* = 0.99). PT levels in the ETRJ50 (*p* = 0.018) and ETRJ100 (*p* = 0.001) groups were significantly higher than those in the EAE group. PT levels in the ETRJ50 (*p* = 0.03) and ETRJ100 (*p* = 0.001) groups were significantly higher than those in the RJ50 group; also, PT levels in the ETRJ100 group were significantly higher than those in the ET group (*p* = 0.02) (Figure 3).

CB1R levels in the EAE group were significantly lower than those in the HC group (*p* = 0.001). There was no significant difference between the sham and EAE groups (*p* = 0.89); however, the levels in the RJ100 (*p* = 0.007), ETRJ50 (*p* = 0.003), ETRJ100 (*p* = 0.03), as well as in the RJ100 (*p* = 0.001), ETRJ50 (*p* = 0.001), and ETRJ100 (*p* = 0.0019) groups were significantly higher than those in the RJ50 group. Additionally, in the RJ100 (*p* = 0.02) and ETRJ50 (*p* = 0.008) groups, the PT levels were significantly higher than those in the ET group (Figure 4).

## 4. Discussion

The results of the present study showed that endurance training had no significant effect on PT and hippocampal CB1R in rats with EAE. Endurance training and RJ separately or in combination could not increase PT to a level similar to that in the HC group. However, ET + RJ50 and ET + RJ100 increased PT compared to that in the EAE group. Regarding CB1R, RJ100, ET + RJ50, and ET + RJ100 increased CB1R compared to that in other EAE-induced groups and to a level similar to that in the HC group. 

Studies have pointed to the favorable role of exercise in improving cannabinoid related pathways in reducing psychological disorders in people with neurodegenerative diseases [20,26]. Researchers believe that aerobic or endurance training increases anandamides in cerebrospinal fluid and induce anti-inflammatory effects in the central nervous system, increasing CB1R and CB2R expression in both blood and brain and spinal cord tissue in EAE and MS patients through improving the function of the GABAergic system, improving neurotrophins, and improving cannabinoid analogues and antagonists [20]. However, peripheral pain in patients with autoimmunity is multifaceted and different mechanisms are effective on it. So, its related mechanisms must be investigated from different perspectives, including hyperactivity of pain sensors in the peripheral organs to physiological and pathological damage to peripheral nerves [32]. In addition to impaired neurotrophins, endorphins [32] and cannabinoids are other mechanisms involved in peripheral pain, of which impaired glutamate and glutamate receptors, impaired expression of inotropic, increased activity and expression of α-amino-3-hydroxy-5-methyl-4-isoxazolepropionic acid (AMPA) and N-methyl-D-aspartate (NMDA), and tumor necrosis factor α (TNFα) and IL-1β can be noted [33].

Regarding exercise, it seems that exercise increases the expression of pain-related genes such as SMAD3 and ASPN through increasing reactive oxygen species (ROS) in the initial hours; this process, which in turn, inhibits myostatin expression and a stimulates growth hormones and protein synthesis, ultimately exerting analgesic effects on the peripheral and lower extremities [34].

Intensity and duration of training seem to be important factors in diagnosing pain, because a review study proposed that high-intensity endurance training leads to increased creatine kinase, C-reactive protein (CRP), and inflammatory cytokines in some athletes, which do not return to the normal state for 28 h [35]. Therefore, it seems that the increase in pain threshold and CB1R in hippocampal tissue of just-exercise groups were not significant because exercise can increase cannabinoid agonists and their analogues, but it also increases cortisol hormone levels, which among other effects, can affect pain [20]. Therefore, despite the limited information regarding the effect of exercise training on cannabinoid receptor in patients with neurodegenerative disorders, it seems that the non-significant increase in CB1R and pain threshold levels in this study is probably related to the type and intensity of exercise as well as other unknown cellular mechanisms in an EAE model.

In this regard, the study of Hosseini et al. showed that considering the effect of exercise training on positive and negative slopes, only training on a positive slope increased pain threshold, while training on a negative slope did not have a significant effect on pain threshold [26]. On the other hand, it appears that the differences in the type of training and adaptation due to the cellular redox pathway in the mentioned study increased motor balance in Alzheimer’s rats, which confirms the results of this study [26].

Our findings showed that RJ50 and RJ100 had no significant effect on PT; however, RJ100 increased CB1R gene expression in the hippocampal tissue of rats with EAE. It is believed that RJ, depending on the plant the bee consumes, contains isoflavones, phenolic acid, 10-hydroquanoic acid, estrogen, and some vitamins that can initially have neurotrophic effects through cerebral, hepatic, and intestinal absorption pathways [18]. In addition, RJ seems to stimulate the expression of antioxidants and increase their activity, and by improving the function of T cells, helps the immune system to reduce inflammatory factors [36]. Thus, the analgesic effects of RJ can be attributed to modulating sympathetic tone, improving neurotransmitters such as serotonin, dopamine, increasing neurotrophins, improving the function of the antioxidant system, and reducing inflammatory factors [18,26,36]. Consistent with the present study, the results of Hosseini et al.’s study showed that consumption of 100 mg/kg RJ had no significant effect on increasing the pain threshold in Alzheimer’s rats but improved motor balance [26]. 

In this regard, previous findings showed that consumption of 200 mg/kg RJ reduced pain perception and pain intensity, and improved neurotrophins in rats with endometriosis [36]. The results of that study are not consistent with the present study, as the difference in the type of damage to the nervous system and the dose of consumed RJ can be two reasons for contradictions in findings.

Because in EAE induction, most tissues of the peripheral and central nervous system are involved in inflammation and sensitivity, and in the present study, a higher doses of RJ caused an increase in CB1R expression, it seems that high doses of RJ have more favorable effects on improving nervous system function. In line with this finding, the studies showed that RJ increased neurotrophins and improved spatial learning and memory in Alzheimer’s rats [37]. Studies also showed that 3 months of RJ treatment substantially ameliorated behavioral deficits of Alzheimer’s disease and its pathology mechanisms [38]. Thus, RJ seems to have lesser analgesic effects at doses lower than 100 mg/kg. 

The present study findings also showed that ETRJ50 and ETRJ100 had a significant effect on the increase of PT and CB1R in the hippocampal tissue of rats with EAE. Regarding the simultaneous effect of training and royal jelly supplementation on the nervous system, the researchers showed that the interaction of training on a positive slope with RJ improved the pain threshold. Findings of another study contradicted present findings which may be due to the nature and mechanism of exercise and eccentric contraction, as training on a negative slope with royal jelly had no significant effect on pain threshold [26]. According to another study findings, RJ treatment can cause mitochondrial adaptation with improved endurance training and some related signaling factors in the soleus muscles of mice [39].

In addition, given the results of the present study indicated the prominent effect of combined training and RJ on the increase of PT and CB1R compared to RJ50 and ET, and also the increase of CB1R gene expression which was dose dependent on RJ, it appears that according to previous findings, exercise training increases CB1R and CB2R expression [20] from the cannabinoid improvement pathway, GABAergic system function, neurotrophins, cannabinoid analogues and antagonists, and increased magnesium anandamides in the spinal cord as well as anti-inflammatory effects, and so manifests its analgesic effects [20,26].

Partial findings of this study indicated that RJ and ET controlled and reduced weight compared to that in the HC group. Considering the fact that overweight increase the risk and symptoms of MS [40], controlling weight was another mechanism related to improving weight in all RJ and ET (separately or in combination) treatment groups. It is believed that RJ has analgesic effects through the neurotrophic pathway [18], increased expression of antioxidants, improved function of immune T cells, reduced inflammatory factors [36], modulated sympathetic tone, improved neurotransmitters such as serotonin and dopamine, and increased neurotrophins [18,26,36]. However, due to the various pain mechanisms in the peripheral and signaling pathways to the central nervous system, the lack of study of inflammatory pathways, AMPA, and NMDA seem to be limitations of the present study and hence assessing these mechanisms and indices are suggested for future studies. In addition, given the non-significant effect of exercise training on the study variables, future studies with longer duration or higher intensities or different modes are recommended.

## 5. Conclusions

Aerobic training and royal jelly consumption alone indicated no significant effect on improving pain threshold levels or CB1R, and the interaction of training and RJ consumption seems to be particularly favorable in the pain-related physiological cannabinoid pathway in the EAE model. Thus, the simultaneous consumption of royal jelly at doses of 50 or 100 mg/kg along with exercise training is recommended to improve PT and related pathways in neurodegenerative diseases.

## Figures and Tables

**Figure 1 nutrients-14-04119-f001:**
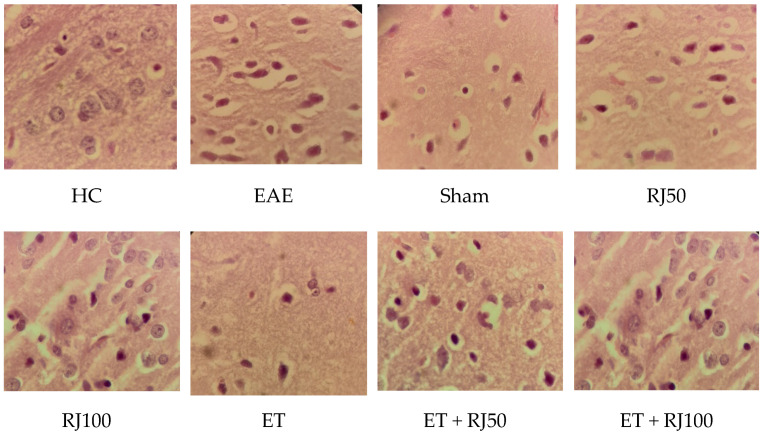
Histopathological analyses of hippocampus. Healthy control (HC); experimental autoimmune encephalomyelitis (EAE); 50 mg/kg royal jelly (RJ50); 100 mg/kg royal jelly J (RJ100); exercise training (ET); exercise training + 50 mg/kg royal jelly (ET + RJ50); exercise training + 100 mg/kg royal jelly (ET + RJ100).

**Figure 2 nutrients-14-04119-f002:**
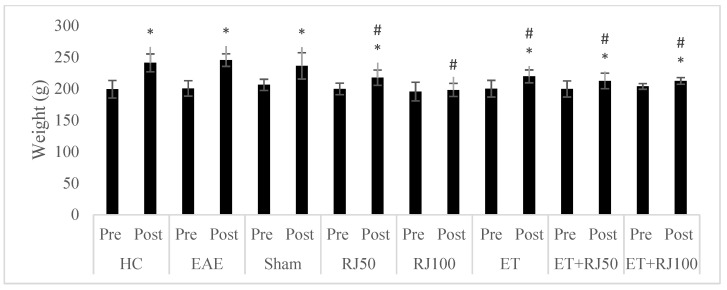
Comparison of weight pre- and posttreatment and with control group. * Significant difference with pretreatment. # Significant difference with HC. Healthy control (HC); experimental autoimmune encephalomyelitis (EAE); 50 mg/kg royal jelly (RJ50); 100 mg/kg royal jelly J (RJ100); exercise training (ET); exercise training + 50 mg/kg royal jelly (ET + RJ50); exercise training + 100 mg/kg royal jelly (ET + RJ100).

**Figure 3 nutrients-14-04119-f003:**
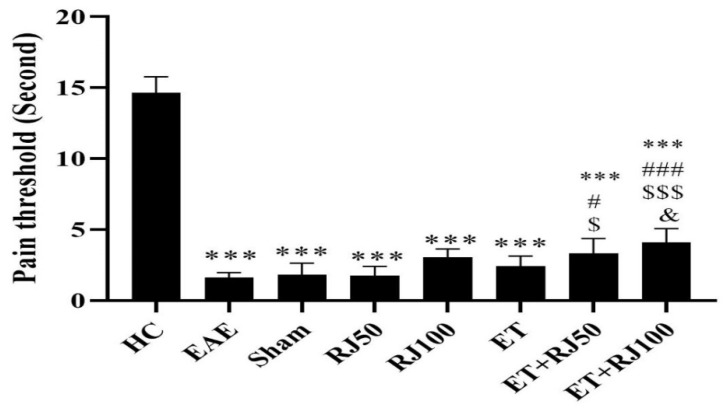
PT levels in rats in the groups of study. EAE reduced PT in all EAE-induced groups, while ET + RT50 and ET + RJ100 augmented PT. Each value is the mean ± SD. *** (*p* ≤ 0.001): significant decrease compared to the HC group. # (*p* ≤ 0.05) and ### (*p* ≤ 0.001): significant increase compared to the EAE group. $ (*p* ≤ 0.05) and $$$ (*p* ≤ 0.001): significant increase compared to the RJ50 group. & (*p* ≤ 0.05): Significant increase compared to the ET group. Healthy control (HC); experimental autoimmune encephalomyelitis (EAE); 50 mg/kg royal jelly (RJ50); 100 mg/kg royal jelly J (RJ100); exercise training (ET); exercise training + 50 mg/kg royal jelly (ET + RJ50); exercise training + 100 mg/kg royal jelly (ET + RJ100).

**Figure 4 nutrients-14-04119-f004:**
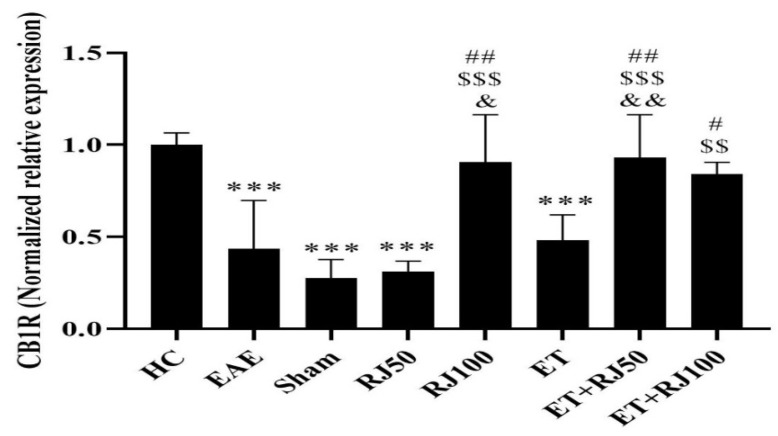
CB1R gene expression levels in the hippocampal tissue of rats in the research groups. EAE reduced CB1R, while RJ100, ET + RJ50, and ET + RJ100 increased CB1R. Each value is the mean ± SD. *** (*p* ≤ 0.001): significant decrease compared to the HC group. # (*p* ≤ 0.05), ## (*p* ≤ 0.01): significant increase compared to the EAE group. $$ (*p* ≤ 0.01) and $$$ (*p* ≤ 0.001): significant increase compared to the RJ50 group. & (*p* ≤ 0.05) and && (*p* ≤ 0.01): significant increase compared to the ET group. Healthy control (HC); experimental autoimmune encephalomyelitis (EAE); 50 mg/kg royal jelly (RJ50); 100 mg/kg royal jelly J (RJ100); exercise training (ET); exercise training + 50 mg/kg royal jelly (ET + RJ50); exercise training + 100 mg/kg royal jelly (ET + RJ100).

**Table 1 nutrients-14-04119-t001:** Sequence of the primers of the research variables.

Gene Name	Sequence of Primers	Product Size (bp)
TBP	Forward: 5′-GCGGGGTCATGAAATCCAGT-3′	147
	Reverse: 5′-AGTGATGTGGGGACAAAACGA-3′	
CB1 receptor	Forward: 5′-AGAACTTACTGTGAACAGGCTCT-3′	105
	Reverse: 5′-ACTGAGAAAGAGATGCCGGG-3′	

## Data Availability

Data are available from the corresponding authors upon request.

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
