# Peer review of "The Effect of Exercise Training and Royal Jelly on Hippocampal Cannabinoid-1-Receptors and Pain Threshold in Experimental Autoimmune Encephalomyelitis in Rats as Animal Model of Multiple Sclerosis"

_nutrients, 2022, doi:10.3390/nu14194119_

Round 1

Reviewer 1 Report

The finding in this manuscript is very interesting, but it needs to be improved.

1.     It would be great to show the EAE disease severity (disease score), or to evaluate if there is any correlation between EAE severity and exercise training and royal jelly on hippocampal 2 cannabinoid-1-receptors.

2.     The figure legends need to be improved. It did not describe the experiment. 

Author Response

We thank Reviewer #1 for their helpful and valuable comments, which helped us to improve the quality of the revision. Please find the detailed point-by-point-response attached as a separate file. 

Thank you again for all your kind efforts. 

Reviewer 2 Report

Although the subject is interesting, the article does not present enough results to support the publication. In addition, the authors use an EAE model not widely used today. The problem is not the experimental model per se, which was described by Feurer et al., 1985 (DOI: 10.1016/0165-5728(85)90006-2) and showed the clinical signs as paralysis. The problem is that the authors did not show the clinical signs or histology of the central nervous system to confirm the disease induction and also the protection mediated by the proposed treatment. It a very simplified presentation of a complex model. 

Author Response

We thank Reviewer #2 for their helpful and valuable comments, which helped us to improve the quality of the revision. Please find the detailed point-by-point-response attached as a separate file. 

Thank you again for all your kind efforts. 

Round 2

Reviewer 2 Report

The authors present a list of articles that use the EAE model but did not include the clinical score and body weight of the sick and sick-treated animals, nor histopathological analyses that prove that the EAE model worked in their hands.

Methods:

"It should be noted that the disease scale was set as follows: Zero: no disease, 1: tail movement disorder, 2: tail paralysis, 3: gait disorder, 4: one- 134 leg paralysis, 5: two-leg paralysis, 6: hands and legs paralysis, and 7: death [27];[28]. In addition, given to the research requirement for animals' minimal daily activities, rats in 2-scales were included and rats on scales 1, 6 and 7 were typically excluded from the study."

Discussion:

"However, due to the various pain mechanisms in the peripheral and signaling pathways to the central nervous system, the lack of study of inflammatory pathways, AMPA and NMDA as well histology of nervous system and EAE disease score seem to be limitations of the present study and hence assessing these mechanisms and indices are suggested for future studies."

In the methods session there is a description of the clinical score saying that it was evaluated. Why isn't that data presented? Why did the authors put in the discussion that they did not do if into the methods?

Author Response

Again, we thank Reviewer #2 for their valuable comments and suggestions. Please find attached the detailed point-by-point-response.

Sincerely

Round 3

Reviewer 2 Report

The authors included histology and body weight analyses of the animals, leaving the results more robust.